# Regional Inequality of Higher Education Development in China: Comprehensive Evaluation and Geographical Representation

**Yong Han** [1], **Ruixing Ni** [1] **and Junbo Gao** [2,*]

1   School of Geographic Sciences, Institute of Higher Education, Xinyang Normal University, Xinyang 464000, China
2   School of Tourism, Xinyang Normal University, Xinyang 464000, China
*   Correspondence: gaojb@igsnrr.ac.cn

**Abstract:** China's higher education has entered the stage of a universal system, enrolling over 50% of students, and the Chinese government takes its high-quality oriented development as its educational goal. However, as the largest developing country in the world, the regional inequality is still a major obstacle to the equitable development of higher education in China. With the 1270 higher education institutions listed by the Chinese Ministry of Education in 2021 as the research objects, the degree of regional inequality of higher education development (HED), and its influencing factors, was visualized after being calculated by geo-statistical methods, such as geographical detectors. The results show that, first, HED in China shows regional linkage and hierarchical connection in high-value regions in the input and outcome dimensions. Second, the input dimension is still the leading factor restricting current HED in China. At the local scale, the geographical stratification characteristics of the four dimensions are evident. The restriction of the educational process covers a wide range and is concentrated in areas with the high-value regions of HED. The innovation of the research is the analysis of the geographical stratification of mechanisms, which identifies regional differences in the factors affecting HED in China.

**Keywords:** regional inequality; geographical representation; higher education development; China

## 1. Introduction

With the expansion of the higher education system in China since the late 1990s, questions on the distribution of higher education opportunities and resources have attracted increasing attention from academics, policymakers, and the general public [1,2]. The renewed attention to this problem in this study stems from the fact that higher education in China has entered a new stage. In 2019, the gross enrollment rate of higher education in China exceeded 50% for the first time, and higher education has entered the primary stage of popularization. The high-quality transformation and development of higher education have become the main goal in this stage. That is, higher education in China has moved from the extension development stage of quantity and scale, into the connotation development stage characterized by quality [3]. Therefore, the regional heterogeneity description of higher education resource allocation in China under the scale orientation can no longer meet our demand for high-quality orientation on higher education development (HED) in China.

The experience of the Industrial Revolution, and the scientific and technological revolution, make us aware that high-quality talents have a profound impact on the promotion of national development, and that a well-educated population is the driving force for a country's sustainable development. Higher education plays a key role in fostering talent. In the face of the transformation and development of higher education in China, the regional imbalance has become an important factor restricting the development of high-quality

higher education in China. The higher education imbalance between regions is not only a problem within education, but also a problem relating to economic and social incoordination in different regions. We need to make it clear that every region has the right to development and should have opportunities for development. Against this background, China needs good higher education to help build an adequately large, qualified workforce. Access to higher education needs to be more equally provided across different provinces in China. Therefore, the cognition of the geographical rules determining higher education resources at the national scale should identify all aspects, from the simple description of the spatial distribution of colleges and universities, to the exploration of the unbalanced development of high quality in higher education.

The remainder of the paper is organized as follows. A brief literature review is provided in the second section. Building on the previous literature, our research concept, including the methodology applied and data used for the analysis, are introduced in the third section. The results of the analysis are presented in the fourth section, and the fifth section is the discussion and conclusions.

## 2. Literature Review

Geographies of education have been burgeoning in the 21st century [4–6]. Although there is no unified definition of the geography of education, its research theme points to geography-related problems in education. Space, place, and scale are the key geography words involved in education. The unbalanced regional allocation of educational resources and the regional imbalance of educational development are the main issues in the geography of education research at the macro scale [7]. The analysis of the spatial heterogeneity of educational resources enables the spatial representation of the unbalanced development of education, providing a scientific basis for the balanced allocation of educational resources at different spatial scales. With the change of the principal contradiction in Chinese society, the unbalanced and inadequate supply of higher education from the spatial perspective shows a large-scale difference. For example, there are significant regional differences in the numbers, quality, structure, and efficiency of higher education resources on the national scale. Geographic factors have gradually become an essential element of resource allocation, whereby geographical origin is highlighted as an important stratifying dimension in access to the opportunity structure. Spatial factors have been increasingly acknowledged as a potential barrier to access to, and subsequent participation [8] in, higher education. Therefore, the spatial exploration of educational resource allocation at the national or regional scale is a scientific starting point for solving the problem of the equality of educational opportunities and outcomes between regions.

Recent research on regional heterogeneity of HED has focused on the spatial pattern description and regularity exploration of the input dimensions [9]. After more than 40 years of HED since the reform and opening up, the spatial allocation of higher education resources in China has gone through three stages: spontaneous allocation, conscious allocation, and the pursuit of rational allocation [10]. At present, China's higher education still shows an imbalance of educational resources between the eastern, central, and western regions. The specific performances are: (1) The distribution of universities and colleges presents the spatial pattern of "more in the southeast, less in the northwest", and "dispersion from national scale and gathering in particular parts of regions" [11–13]. (2) The spatial distribution of different types of universities varies. For example, the spatial distribution of public universities and undergraduate universities is roughly the same. Many private universities and cooperative universities are located in southeast coastal provinces. The spatial distribution of specialized colleges is wide, and the distribution between provinces is relatively balanced. "Double first-class" construction universities are highly concentrated in megacities and urban agglomerations. (3) From the perspective of the structural characteristics of the spatial distribution, China's higher education resources are mainly concentrated in the provincial capitals of municipalities directly under the central government and provinces, and have a multi-center structure distribution [14]. At present, the

spatial structure of China's HED has not been improved, and the goal of the coordinated regional development of higher education has not been fully realized [15]. (4) There are also huge differences in the size of universities as represented by enrollment and faculty numbers. Since 1949, although the number of college students in each province has been expanding, higher education is still mainly concentrated in the east, or in regions with high population density. The absolute gap in higher education quantity among provinces is also increasing, and the degree of dispersion fluctuates [16]. Simultaneously, the increasingly unequal distribution of higher education resources among students was found to be even more severe when considering teaching resources measured by the number of computers, by universities' educational expenditure, or by the size of the universities' fixed assets, again in favor of the economically more advanced regions [17]. For example, the Beijing–Tianjin–Hebei, Yangtze River Delta, and Pearl River Delta regions all show a single center form, and the spatial agglomeration degree of full-time teachers is higher than that of school students [9]. The main reasons behind the current situation are that, since the reform and opening up, regional inequality continues during China's transition to a market economy. The development inertia of regional inequality in educational input, the college entrance examination system, and the scattered enrollment plan system, triggered an unequal opportunity structure, presenting spatial imbalance characteristics at different scales [18–20]. With the transformation of China's HED from the elitism stage to the popularization stage, the teaching quality, regional economic development level, local cultural level, and the distance of universities from the source of students have gradually become important factors affecting regional differences in higher education resource allocation and the interconnection among types of higher education [21].

In short, because China is the largest developing country in the world, its HED has been attracting academic attention at home and abroad. There have been fruitful results in research on the spatial distribution and regularity of higher education resource allocation. However, it is still meaningful to engage in research on HED in China at the national scale and in its spatial dimensions. The significance of this research is based on the following two prerequisites: The background of China's HED has moved into the primary popularization stage. The development of higher education in China has entered a new stage from the extension development oriented by quantity to the connotation development characterized by quality. High-quality and connotative development has become the goal guiding the development of higher education. Overcoming the trap of the regional Matthew effect, and promoting regional balanced development, are real issues to be addressed in realizing the high-quality development of higher education. Conversely, due to its vast territory and special geographical features, China has a long history of uneven regional distribution of resources and population. Therefore, the cognition of scientific rules, such as the regional division and regional difference characteristics of influencing factors, is the scientific starting point in clarifying the balanced regional development of higher education in the popularization stage.

Therefore, based on the changes in the background, our study holds that the spatial dimensions of higher education at the national scale are worthy of further research in the following three respects: First, previous studies focused primarily on the geographical distribution of educational resources, and funding, and the implications of this for educational attainment and skills outcomes. They only considered the geographic imbalance in higher education in a single dimension. With the transformation of China's higher education to connotative development, the high-quality dimension should be regarded as a development goal throughout the whole system of higher education. While an increasing number of studies have focused on the development of higher education opportunity equality in China, quantitative, systematic research on the distribution of higher education resources across China is still rather limited. Second, the interpretation of the influencing factors mainly focused on the external system of higher education for the unbalanced allocation of educational resources, with the perspective at the national scale. This approach ignores the internal system of higher education, and the regional differences of different influencing

factors at the local scale. Therefore, we hypothesize that there must be geographical differences in the driving mechanisms behind the imbalanced characteristics of HED. Third, the intervention of geographical statistical methods, and the supplement of multiple data, needs to be added to provide technical support for solving the above problems. For example, research methods such as geographical statistical analysis have been involved in the research field of regional differences in higher education [22], but they are still relatively limited. In particular, the characteristics of regional mechanism differences that affect the allocation of higher education resources remain under-explained. In terms of data, official statistics based on provincial or municipal scales were the main sources used to demonstrate the above content. This macro-data is too general, and the limited content of the relevant data cannot reflect the whole meaning of connotative development [23]. There is a need for supplementary data based on the individual level of schools.

Based on this, taking the 1270 higher education institutions listed by the China Ministry of Education in 2021 as research subjects, the evaluation index system of HED from the four dimensions of educational input, access, process, and outcome is constructed. Further analysis of the regional differences in HED is made using a geographic detector and geographic-weighted regression. We provide evidence of how regional inequalities intersect with educational input, access, process, and outcomes in contemporary China, and provide a scientific basis for the practical issues of regional balanced allocation of higher education resources.

## 3. Materials and Methods

### 3.1. Index System

Different purposes give rise to different views on the evaluation of regional education development. Since the 1970s, with the development of education internationalization, the OECD, UNESCO, the WB, and country-level organizations have successively launched their own relatively independent education index systems to measure and compare education development in various regions. For example, the OECD education development index system is based on the "input–output" mode of economics, forming the analysis mode of "background–input–process–output". The education development indicators are correspondingly counted and described from the four dimensions of educational background, educational input, educational process and educational output.

Taking the World Bank's index system of educational development as a reference, our study selects the four indicators of educational input, educational access, educational process, and educational outcome for evaluation from the supply level (Table 1). Education input is the basis and guarantee for the development of all educational activities. It is one of the important indicators used to measure the priority development of education in a region. Its most direct embodiment is the input of financial funds in education, the number of enrollments, the number of teachers, and the number of schools.

When inequality of opportunity is discussed in higher education, it typically pertains to access to college [24]. Under the background of selective admission, the admission opportunity is the result of the matching of students' abilities with university teaching quality, the direct embodiment of a coupling relationship between the number of candidates (regional difference in demand) and the number of enrollments (regional difference in supply). Therefore, the comprehensive results of the first choice and the admission of students are an objective reflection of the above problems.

As a special practical activity, the educational process is the key link to ensuring educational efficiency. In theory, the teaching process is the result of the interaction between teachers and the institution, and it is the common performance of both subjects. Based on the three basic functions of university teaching, scientific research, and local service, the teaching quality and scientific research quality indicators are selected to reflect the higher education process.

Educational outcome is the direct embodiment of the performance of higher education. It is the comprehensive assessment of the implementation effect of the education starting

point and the educational process, the standard used to measure the quality of education and the ultimate goal of educational equality.

**Table 1.** Indicators of the Higher Education Development Index in China.

| Level 1 Indicators | Secondary Indicators | Indicator Instructions |
| --- | --- | --- |
| Input | Enrollment Number ($X_1$) | Annual enrollment number of higher education in the region |
| | Teacher Number ($X_2$) | The number of teachers in higher education in the region |
| | University's Number ($X_3$) | The number of colleges and universities in the region |
| Access | Priority Selection ($X_4$) | The standard of performance in an examination allowing students to select their university |
| | Quality of Admission ($X_5$) | The quality of the comprehensive scores of the students admitted by colleges and universities |
| Process | Talent Cultivation ($X_6$) | The ability of colleges and universities to train talents |
| | Comprehensive Level of Teachers ($X_7$) | Teaching and scientific research level of teachers in colleges or universities |
| | Scientific Research ($X_8$) | Comprehensive scientific research ability of colleges or universities |
| | Teachers Performance ($X_9$) | Performance level of teacher education in colleges or universities |
| Outcome | Employment Quality ($X_{10}$) | The quality of the employment units for graduates |
| | Enrollment Rate ($X_{11}$) | Ratio of continuing education for postgraduates |

Based on the above discussion, a multi-dimensional evaluation index system is constructed to express HED in China. The formula is as follows:

$$\text{HED} = f(X_I, X_A, X_P, X_O) \tag{1}$$

*3.2. Main Methods*

3.2.1. Entropy Weight TOPSIS

The entropy weight TOPSIS method is essentially an improvement of the traditional TOPSIS evaluation method. The weight of the evaluation index is determined through the entropy weight method, and then the TOPSIS method uses technology to approximate the ideal solution. The entropy weight method is used to objectively determine the weight according to the information provided by each evaluation index. As the weight number, it can objectively reflect the importance of an index in the index system when making decisions. Due to the length of this study, the detailed formula operation can be found in Martin's article [25].

3.2.2. Geographic Detector

The geographic detector is a statistical model for spatial data analysis, used for detecting spatial heterogeneity, along with a set of statistical methods that reveal the driving force behind it [26]. The core idea is based on the assumption that if an independent variable has an important impact on a dependent variable, then the spatial distribution of the independent and dependent variables should be similar. Here, factor detection and interaction detection in geographic detectors are used to analyze the spatial distribution of HED in China.

Difference and factor detection is used to detect the spatial differentiation of the local HED level (Y); and, to the extent that it can detect a variable X, explains the spatial differentiation of attribute Y. Using the *q*-value measurement, the expression is:

$$q = 1 - \frac{\sum_{h=1}^{L} N_h \sigma_h^2}{N\sigma^2} = 1 - \frac{SSW}{SST} \tag{2}$$

$$SSW = \sum_{h=1}^{L} N_h \sigma_h^2 \tag{3}$$

$$SST = N\sigma^2 \tag{4}$$

where: $h = 1, \ldots, L$ is the strata of variable Y or factor X, namely, classification, or partition. $Nh$ and $N$ are the number of cells in layer $h$ and the full region, respectively; $\sigma_h^2$ and $\sigma^2$ are the variance of the Y values of layer $h$ and the region, respectively. $SSW$ and $SST$ are within the sum of the squares and the total sum of the squares, respectively. The value domain of $q$ is [0, 1], and the larger the value is, the more obvious the spatial heterogeneity of Y. If the stratification is generated by the independent variable X, the larger the $q$-value, the more it indicates that the independent variable X explains attribute Y, and the weaker the other is. In the extreme case, a $q$-value equal to 1 indicates that variable X completely controls the spatial distribution of Y, whereas a $q$-value equal to 0 indicates that variable X has nothing to do with Y, and a $q$-value indicates that X explains $100 \times q\%$ of Y.

Interaction detection is used to identify the interaction between different risk variables. For example, whether variable $X_1$ and $X_2$ combined increase or weaken the explanatory force on the dependent variable Y, or whether the effects of these factors on Y are independent of each other. The evaluation was performed by first calculating the two factors $X_1$ and $X_2$ separately. The $q$-value for Y: $q(X_1)$ and $q(X_2)$, and calculate the $q$-value they generate when interaction $q(X_1 \cap X_2)$. The relationship between the factors can be divided into the following types (Table 2).

**Table 2.** Types of interaction detectors.

| Criterion | Interaction |
|---|---|
| $q(X_1 \cap X_2) < \mathrm{Min}[q(X_1), q(X_2)]$ | Nonlinear attenuated |
| $\mathrm{Min}[q(X_1), q(X_2)] < q(X_1 \cap X_2) < \mathrm{Max}[q(X_1), q(X_2)]$ | Single variable non-linearity attenuated |
| $q(X_1 \cap X_2) > \mathrm{Max}[q(X_1), q(X_2)]$ | Double variables enhancement |
| $q(X_1 \cap X_2) = q(X_1) + q(X_2)$ | Independence |
| $q(X_1 \cap X_2) > q(X_1) + q(X_2)$ | Nonlinear enhancement |

3.2.3. Geographic-Weighted Regression

Geographic-weighted regression (GWR) is an improved spatial linear regression method that connects the related variables in the space with the regression principle, using local features as the weight reference. The regression coefficients of the GWR are not spatially global or invariant, allowing them to vary somewhat in the spatial regions. The GWR method can consider multiple influence variables simultaneously, and conduct the regression coefficient estimation with the nearest neighboring subsample, directly, effectively describing the spatial non-stationary relationship between the variables. The GWR model can be expressed as follows:

$$y_i = \beta_o(u_i, v_i) + \sum_{i=1}^{k} \beta_k(u_i, v_i)x_{ij} + \varepsilon_j \tag{5}$$

where: $y$ is the dependent variable; $x$ is the independent variable; $(u_i, v_i)$ are the coordinates of the sample $i$; $\beta_0(u_i, v_i)$ is the intercept item; $\beta_k(u_i, v_i)$ is the regression coefficient $k$ on the sample $i$, which is a function of geographic location; and $\varepsilon_i$ is the regression residuals.

The regression coefficients of the GWR model are estimated by the weighted least squares method:

$$\hat{\beta}(u_i, v_i) = \left[ X^T W(u_i, v_i) X \right]^{-1} X^T W(u_i, v_i) Y \tag{6}$$

where: $W(u_i, v_i)$ is the spatial weight diagonal matrix; $X$ is the independent variable matrix; and $Y$ is the dependent variable vector.

The spatial weights are calculated using the Bi-square function:

$$\begin{cases} W_{ij} = \left[ 1 - \left( d_{ij}/h \right)^2 \right]^2, \, d_{ij} < h \\ \qquad W_{ij} = 0, \, d_{ij} \geq h \end{cases} \tag{7}$$

where: $w_{ij}$ is the weight when the spatial known point; $j$ goes to estimate the unknown point $i$; $d_{ij}$ is the Euclidean distance between points $i$ and $j$; and $h$ is the bandwidth.

Bandwidth is judged by the minimum *AICc* information criterion (corrected Akaike information criterion):

$$AICc = -nln\left( \sigma' \right) + nln(2\pi) + n\{ [n + tr(S)]/[n - 2 - tr(S)] \} \tag{8}$$

where: $n$ is the number of data points; $\sigma'$ is the error term (estimated standard deviation); and $tr(S)$ is the trace of the hat matrix $S$.

### 3.3. Data Resources and Processing

The 1270 higher education institutions (undergraduate schools) listed by the China Ministry of Education in 2021 were taken as objects of our study, and the data were averaged to the municipal units, whereby spatial statistical analysis was conducted with the municipal spatial units. The data come from two sources: The first is the data on education input in the 2020 China Statistical Yearbook, The China 2020 Urban Yearbook, and The Statistical Bulletin of Urban National Economic and Social Development in 2020. The second is the data on educational access, educational process, and educational outcomes in *Choosing a University and Select a Major in 2020 (General Universities edition)*. *Choosing a University and Select a Major* is a reference book for college entrance examination applicants, with university ranking, subject category ranking, and major ranking, supplemented by teacher level ranking and graduate quality ranking. The book is produced by the China Statistics Press and Professor Wu Shulian. It has been published continuously for 20 years and has a high degree of influence on the selection of schools in China's college entrance examination.

In our study, the rank correlation coefficient was used to calculate the data. As enrollment number ($X_1$), teacher number ($X_2$), and university number ($X_3$) are statistics, other indicators are the rank data. To maintain consistency, the statistical data were graded for processing. The specific methods are as follows: The overall data is divided into 11 levels from small to large, and the larger the grade score, the higher the original data level. Grades 1 and 2 account for 15% of the overall data; 10% of the overall data are accounted for from Grade 3 to Grade 8; Grade 9 accounts for 5% of the overall data; Grade 10 accounts for 3%; and Grade 11 accounts for 2% of the overall data. Taking Tsinghua University as an example, its raw data shows the number of teachers as 5892, the priority selection A++, the quality of admission A++, the value of talent cultivation 77.16, the comprehensive level of teachers A++, the value of scientific research 68.53, the teachers' performance level A+, the employment quality A++, and the enrollment rate A++. After reclassification, the highest grade of indicator is the eleventh grade, which is given the value of 11, and so on. So, the indicators mentioned above were reclassified and revalued as 11 (indicators of the number of teachers, priority selection level, quality of admission level, etc.) and 10 (indicator of teachers' performance).

### 4. Empirical Results

### 4.1. Regional Heterogeneity of Higher Education Development in China

The indicators were calculated by the entropy weight TOPSIS method, and the data were spatially fit by the nuclear density estimation. The estimated values were expressed visually according to the natural fault point classification method based on ARCGIS Pro (Figure 1). The results show that the high-value areas of China's HED are concentrated in Beijing and Tianjin, Shanghai, Nanjing, and Hangzhou, and Guangzhou in the Pearl River

Delta. Only Wuhan and Xi'an in central China have good higher education performance. Specifically, from the evaluation dimension of HED, the high-value area of higher education input is concentrated in the southeast Yangtze River Delta region and the Pearl River Delta region. Henan and Shandong provinces are the major provinces for the college entrance examination, and where the higher education input is also distributed at high values. In western Chongqing city, the input level of higher education shows a leading trend (Figure 1b). From the perspective of higher education access, only Beijing and Tianjin and the Shanghai–Nanjing–Hangzhou region in the Yangtze River Delta have opportunity advantages (Figure 1c). The spatial pattern characteristics of the educational access are similar to those of the higher education process. This also indicates that the performance of universities in talent training and scientific research level is the objective stimulus of students' school choice (Figure 1d). Finally, according to the results of higher education reflected by the employment and continuing education of graduates, Beijing–Tianjin and Shanghai–Nanjing–Hangzhou serve as the first-level core areas, showing a spatial pattern of gradually weakening to the central and western regions (Figure 1e).

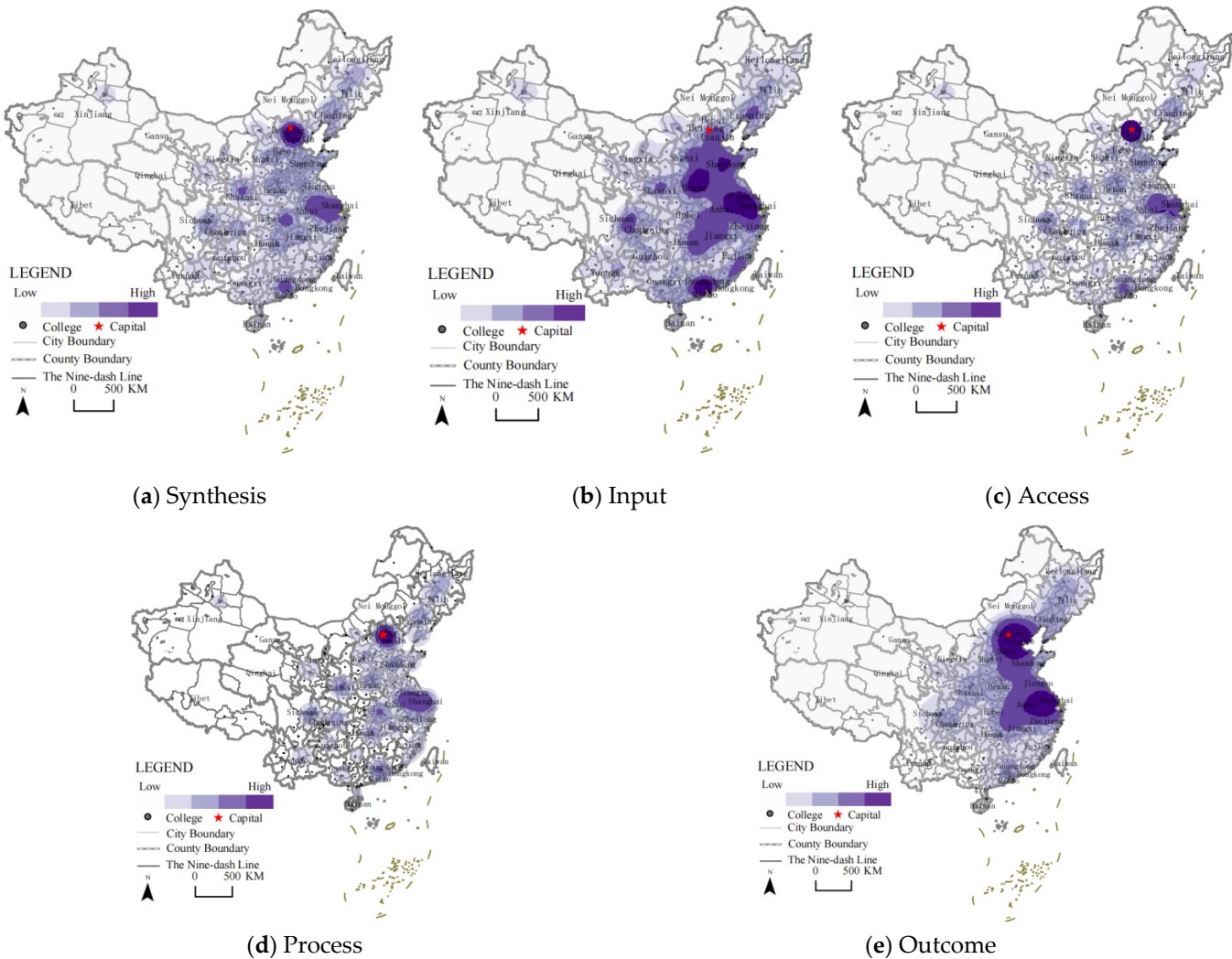

**Figure 1.** The distribution of TOPSIS results of HED in China in 2020.

### 4.2. The Factors Driving the Spatial Differences of HED

#### 4.2.1. Analysis of Leading Factors

In light of the regional heterogeneity of HED in China, the geographical detector method was used to quantitatively analyze the influencing factors. The $C_i$ value of HED was set as the dependent variable Y, and the secondary index $X_1$–$X_{11}$. as the independent

variable. Formulas 2 to 4 were used to calculate and determine the leading factors affecting HED through the $q$-value, and the calculations were tested by $p$ value (Table 3). The results show that, apart from the low values in $X_4$ and $X_9$, the gap between other high-value factors is not large, indicating that there may be a large inter-correlation between the various factors. According to the results, input in higher education is still the leading factor restricting the development of higher education in China. The allocation of the educational resources represented by the number of regional university teachers ($X_2$) and the number of universities ($X_3$) is still the leading factor causing the unbalanced pattern of HED. For the access dimension of higher education, the quality of admitted students ($X_5$) is the main factor affecting the development level of higher education, while priority selection ($X_4$) has little impact on HED. This shows that Chinese universities still take examination grades as the main reference when admitting students, and the students' expected goals are not fully met. There are subjective and objective factors at work here. For the process dimension in higher education, the $q$-value of teachers' performance ($X_9$) is less powerful in explaining the HED level, whereby the performance level of teachers' education in colleges or universities seems to be less related to the level of HED. It also explains the current situation in China of "emphasizing scientific research rather than teaching" .

**Table 3.** Results of factor detector.

| Factor | $X_1$ | $X_2$ | $X_3$ | $X_4$ | $X_5$ | $X_6$ | $X_7$ | $X_8$ | $X_9$ | $X_{10}$ | $X_{11}$ |
|---|---|---|---|---|---|---|---|---|---|---|---|
| $q$ | 0.66 | 0.77 | 0.74 | 0.38 | 0.65 | 0.69 | 0.68 | 0.69 | 0.36 | 0.67 | 0.67 |
| $p$ | 0 | 0 | 0 | 0 | 0 | 0 | 0 | 0 | 0 | 0 | 0 |

Note: The value range of $q$ is [0, 1], a value close to 1, indicates that the factor affects the development level of higher education; whereas a value close to 0 indicates that the influence of the variable is weak. All 11 variables have passed the significance test.

### 4.2.2. Analysis of Interaction Detection

From the perspective of system theory, the change in HED is not the result of the linear effect of a single factor. Meanwhile, from the above analysis, there may be a symbiotic relationship between the regional differences in China's HED. Therefore, the results need to be further tested using interactive detection methods. The calculation results in Table 4 show that all interaction detection between the 11 variables shows double variables enhancement, whereby the influence of the factor explanatory force after the interaction is stronger than the original individual factors, indicating that HED is the result of multi-factorial interaction. However, the analysis results of the above leading factors show that our evaluation index system also reflects the shortcomings of HED in China. Whether these weaknesses can be supplemented in other ways needs further analysis. From the perspective of the interaction results of various independent variables, the maximum value in the matrix appears in the interaction node with the independent variable $X_2$, indicating that the number of university teachers is the basic element necessary to ensure the whole process of HED.

**Table 4.** Interaction detector results.

| Factor | $X_1$ | $X_2$ | $X_3$ | $X_4$ | $X_5$ | $X_6$ | $X_7$ | $X_8$ | $X_9$ | $X_{10}$ | $X_{11}$ |
|---|---|---|---|---|---|---|---|---|---|---|---|
| $X_1$ | 0.66 | | | | | | | | | | |
| $X_2$ | 0.80 | 0.77 | | | | | | | | | |
| $X_3$ | 0.79 | 0.84 | 0.74 | | | | | | | | |
| $X_4$ | 0.88 | 0.90 | 0.88 | 0.38 | | | | | | | |
| $X_5$ | 0.87 | 0.89 | 0.87 | 0.67 | 0.65 | | | | | | |
| $X_6$ | 0.88 | 0.91 | 0.88 | 0.70 | 0.69 | 0.69 | | | | | |
| $X_7$ | 0.88 | 0.91 | 0.88 | 0.69 | 0.69 | 0.70 | 0.68 | | | | |
| $X_8$ | 0.88 | 0.90 | 0.88 | 0.71 | 0.69 | 0.70 | 0.70 | 0.69 | | | |
| $X_9$ | 0.88 | 0.90 | 0.87 | 0.39 | 0.66 | 0.71 | 0.69 | 0.71 | 0.36 | | |
| $X_{10}$ | 0.87 | 0.89 | 0.87 | 0.68 | 0.69 | 0.71 | 0.71 | 0.71 | 0.68 | 0.67 | |
| $X_{11}$ | 0.87 | 0.90 | 0.87 | 0.69 | 0.70 | 0.72 | 0.72 | 0.72 | 0.69 | 0.68 | 0.67 |

### 4.3. Regional Differences of Mechanism for HED

The study of the influencing factors through geographical detectors is based on the national scale. China has a large land mass and abundant resources, and its economic and social development determines the large differences in the regional educational environment. To solve the problem of regional inequality of HED, it is necessary to identify the influencing factors limiting different regions. Therefore, the GWR model was used to calculate and analyze the spatial heterogeneity of the influencing factors on HED in China. By fitting analysis of the $C_i$ values of HED with its four dimensions, regression coefficient values were calculated according to formula 6. After calculation, the residual squares were 0.0045, 0.0002, 0.0105, and 0.019, and $R^2$ was 0.99, 0.97, 0.90, and 0.99, respectively, which indicated that the results of the GWR model have a high degree of fitting with the observed data. Then, five categories, according to the natural break point analysis method, were identified according to the regression coefficient field values. The larger the regression coefficient value, the greater the dependence of the development level of higher education in this region. Meanwhile, the influence factors were divided into five levels based on the values of the regression coefficients, the types and levels of influence factors in each city were identified, and the main types of influence factors in each city were screened (Figure 2). Overall, from the perspective of the maximum regression coefficient of all dimensions, the degree of HED is the most dependent on education input, which is consistent with the results based on geographical detectors at the national scale. From the results of the geographical weighted regression analysis, the constraints of the HED in China show great regional differences.

1. The education process dimension is a main restriction factor that covers most regions in China, while other regions receive multidimensional influence. Regions restricted by the educational process include the central and southern regions in northeast China, north China, central China, east China, eastern south China, and eastern regions in northwest China, which means that improving the cultivation process in such areas can enhance the quality of higher education.

2. Regions jointly restricted by four dimensions include the northern regions in northwest China and the western and central regions in southwest China, which means that such regions have the lowest HED in China, and all four dimensions need to be improved simultaneously to effectively improve the quality of higher education. To investigate the reasons for this, firstly, the land is sparsely populated, and the climate is harsh. The Qinghai–Tibet Plateau, Kunlun Mountains, Inner Mongolia Plateau, and other terrain in this region cause the sparse population, and the climate is not suitable for human activities. Secondly, residents live and work in peace and contentment and are content with the status quo. The rich natural resources, abundant cultivated land resources, and abundant grassland make the local residents mainly engaged in agriculture and animal husbandry, and the Chinese government has a high inclination to agriculture and animal husbandry subsidies, so the farmers and herdsmen live a rich life and lack the motivation to learn to change their fate. Therefore, the regional government should develop higher education in the region where natural conditions are most suitable for human activities. At the same time, the publicity of higher education should be strengthened, so that farmers and herdsmen can realize the advantages of higher education.

3. Regions jointly restricted by three dimensions include the western and central regions in northwest China, the western regions in southwest China, and the northern regions in northeast China, which means that such areas need to increase their investment in higher education, expand the opportunities of higher education, and improve the quality of higher education results. To investigate the reasons for this, the areas are large and sparsely populated, and the climate is harsh. Geographic factors in the Kunlun Mountains, the Taklimakan Desert, and the Tianshan Mountains make the population sparse and the climate unsuitable for human activities. Therefore, the

regional government should develop higher education in the regions where natural conditions are most suitable for human activities.

4. There are complex and different geographical representations in the east region of southwest China, and the west and south regions of south China, which means that the development situation of various cities in the region is complicated. Firstly, Chongqing municipality and its neighboring two cities are restricted by the access dimension, because the population far exceeds the local higher education opportunity. Secondly, educational access is the main restriction in Guizhou, Hainan, Guangxi, and northern Yunnan regions due to poor economic development, rugged terrain, and people's low level of willingness to attend universities. Thirdly, the southern region of Yunnan province has complex terrain and vertical valleys, and the investment cost of higher education far exceeds that in other provinces. In addition, in the neighboring border areas, a well-developed border trade and a high labor demand reduce the local people's willingness to pursue higher education, ultimately resulting in poor education outcomes.

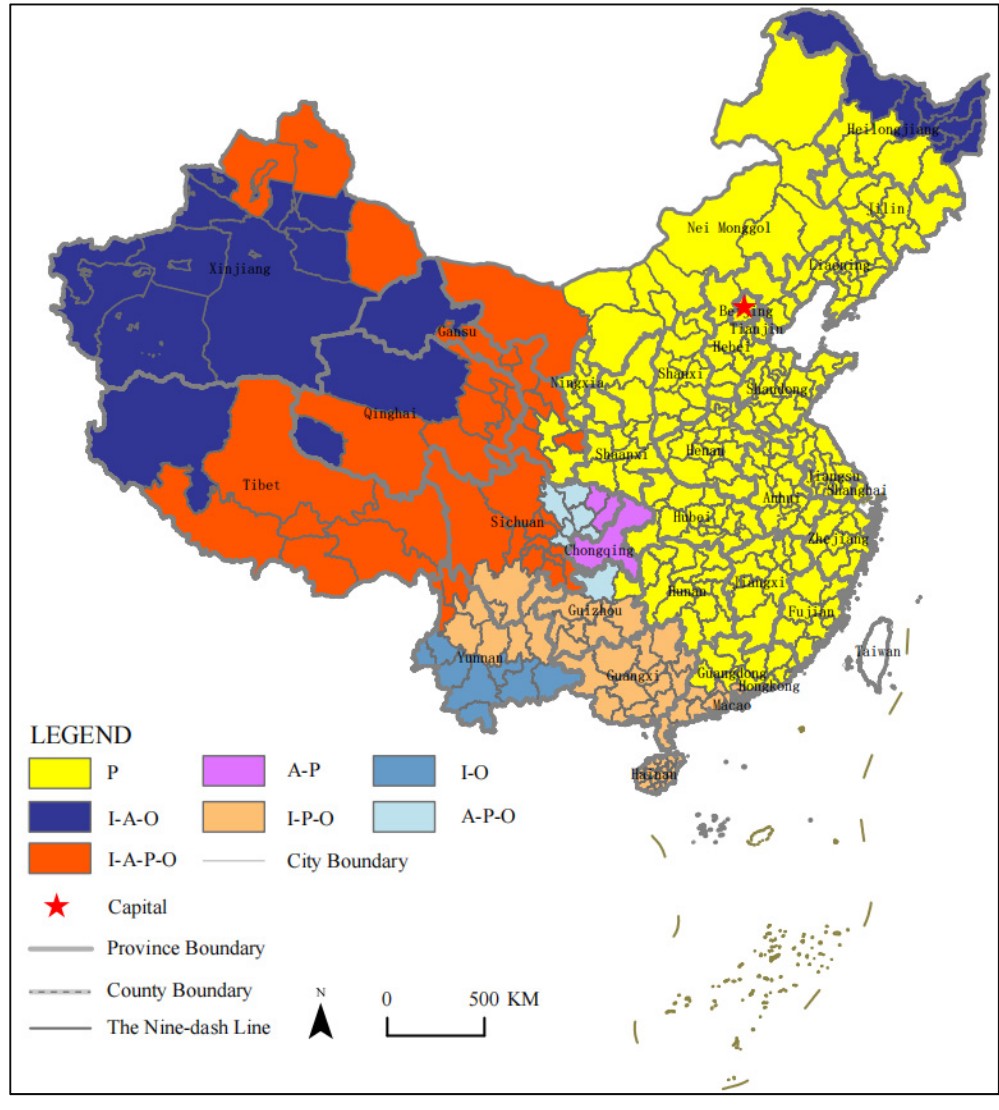

**Figure 2.** Spatial expression of regional inequality of driving forces for HED.

## 5. Discussion and Conclusions

Although many scholars have been involved in researching the regional differences of HED in China, the research scale was mostly confined to provincial and municipal units.

At the same time, the HED evaluation was mainly based on one-dimensional indicators, such as the quantity of universities and teachers or enrollment numbers. Studies using a multi-dimensional comprehensive evaluation were scarce. With the development of higher education in China from the mass stage to the universal stage, it has become a practical issue to improve the quality of HED and realize the regional equitable development of higher education. In this context, our study analyzed the development level of higher education and the regional heterogeneity characteristics of Chinese higher education from four dimensions: educational input, educational access, educational process, and educational outcome.

The spatial agglomeration characteristics of China's HED are evident. There are differences in the quantity between the eastern, central, and western regions, a high-level agglomeration in the Beijing–Tianjin–Hebei region, the Yangtze River Delta, and the Pearl River Delta, and the central characteristics of provincial capitals in the eastern and central regions. Specifically, HED in China shows the characteristics of regional linkage in the educational input and outcome dimensions. Meanwhile, the trend of high-value regions indicates hierarchy and relevance, the double core characteristics in the Beijing–Tianjin–Hebei–Yangtze River Delta line, and its circular structure.

At the national scale, the input dimension is still the dominant one restricting the current HED, which means China is still at the stage of scale expansion in higher education. At the local scale, the geographical stratification characteristics from the four dimensions are evident. The dimensions of input, access, and outcome show a basic trend of gradually increasing from the east-central region to the west and the northeast. However, there is a wide range in the process dimension, most of which overlap with high-value regions of HED, such as the Beijing–Tianjin–Hebei region and the Yangtze River Delta region. The low-value HED regions are the result of multi-dimensional interaction. The research conclusions provide a scientific basis for achieving the balanced development of higher education in China.

Regional inequality in resource allocation, education access, education process, and the quality of higher education outcomes are common HED problems in most countries. The scientific evaluation of the regional inequality of the development level and its influencing factors can provide a scientific basis for achieving balanced regional development. However, it is not enough to discuss the regional imbalance of HED purely from the internal system in higher education. For example, regional equality in higher education needs to consider the relationship between supply and demand. Moreover, the problem of regional imbalance is not simply a problem of spatial distribution. The improvement of the higher education system structure in a specific region is also the basis for realizing the sustainable development of higher education. Therefore, the balanced development between supply and demand needs to take the internal and external system in higher education into consideration, such as disciplines, majors, local industrial structure, and talent demand, which will be the focus of the next step.

**Author Contributions:** Conceptualization, Y.H.; methodology, Y.H.; software, R.N.; validation, Y.H.; formal analysis, R.N.; data curation, R.N.; writing—original draft preparation, Y.H.; writing—review and editing, J.G.; visualization, R.N.; supervision, J.G. All authors have read and agreed to the published version of the manuscript.

**Funding:** This research was funded by National Natural Science Foundation of China (U1904125).

**Data Availability Statement:** Not applicable.

**Conflicts of Interest:** The authors declare no conflict of interest.

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
