# Peer review of "Regional Inequality of Higher Education Development in China: Comprehensive Evaluation and Geographical Representation"

_sustainability, doi:10.3390/su15031824_

Round 1

Reviewer 1 Report (New Reviewer)

Please see attachment below. 

Author Response

Thank you for your recognition of our work, your affirmation is the motivation for our future actions

Section 4.3 has been checked and modified.

All changes have been marked up by using the “Track Changes” function.

Reviewer 2 Report (New Reviewer)

My congratulations to the authors, because they have dealt with an important topic with a methodology rarely covered in academic papers. Their article can serve to replicate similar studies in other regions, and thus be able to generalize the topic covered in their research.

Author Response

Thank you for your recognition of our work, your affirmation is the motivation for our future actions.

Reviewer 3 Report (New Reviewer)

Go through very carefully to clarify technical or jargonistic phrases in the abstract, beginning and end of the paper, in fact go through the whole to see whether definite ('the') and indefinite ('a' or 'an') artricles need to be added

Check whether the documentation method conforms to Sustainability's expectations

Think about the issue of specialist universities (e.g., Xian's Foreign Language University; Sichuan's Universities for Nationalities) for the argument

At some point discuss where access to foreign university experiences affects the analysis, through given universites' ties with the outside world

Otherwise very clever and much needed spatio-geographical work

Perhaps the mathematization is too putatively exhaustive and therefore premature, before specific Chi-square relationships are worked through more patiently?

GWT

Author Response

Point 1: Go through very carefully to clarify technical or jargonistic phrases in the abstract, beginning and end of the paper, in fact go through the whole to see whether definite ('the') and indefinite ('a' or 'an') artricles need to be added 

Response 1: We have gone through the whole manuscript and modified the English expression, especially for technical or jargonistic phrases in the abstract, beginning and the end of the paper. All changes have been marked up by using the “Track Changes” function.

Point 2: Think about the issue of specialist universities (e.g., Xian's Foreign Language University; Sichuan's Universities for Nationalities) for the argument.

Response 2: Our work here focused the development of China's higher education from the macro scale, and did not involve analysis from the micro scale. Research on the micro scale, such as the geographical location of the specialist university and its spatial effect, will be the focus of our next work.

Point 3: At some point discuss where access to foreign university experiences affects the analysis, through given universites' ties with the outside world.

Response 3: Thanks for your advise on the “universities ties with the outside world”. In fact, it deals with another topic of the geography of higher education, namely the mobility of education. Your suggestion is the direction of our future action.

Point 4: Perhaps the mathematization is too putatively exhaustive and therefore premature, before specific Chi-square relationships are worked through more patiently?

Response 4: We have added statistical test in the analysis of geographic detector and GWR model.All changes have been marked up by using the “Track Changes” function.

This manuscript is a resubmission of an earlier submission. The following is a list of the peer review reports and author responses from that submission.

Round 1

Reviewer 1 Report

I accepted this paper despite knowing that the manuscript needs extensive English language editing is needed prior to publication. I see the merits of this paper and the applicability of its methodology in the geographies of education or its spatial phenomenon. The paper recognizes that the Chinese government established "double first-class universities" in major megacities. However, in its discussion, it failed to recognize the fact that the Chinese government is marginalizing peripheral non-Han-dominated provinces due to their distinct cultural differences. If you look at the map of "Double First-Class Universities," as well as Figure 1 in the paper, the distribution of prestige consideration among Chinese universities is primarily condensed to the eastern part of the country (Beijing, Shanghai, Nanjing, Wuhan, Sichuan, among others). That regional imbalance of educational resources mentioned in the manuscript is precisely due to the marginalization of peripheral provinces historically resistant to the Han-dominated Chinese government. There is a need to include such sociopolitical reality in the discussion. 

Author Response

China is a multi-ethnic unified country dominated by the Han nationality with cultural differences between different ethnic groups. However, since the founding of the People's Republic of China, the Communist Party of China and the Chinese government have established and implemented the ethnic policy of ethnic equality, ethnic unity, regional ethnic autonomy, and the common prosperity of all ethnic groups. The development of ethnic higher education and ethnic policy complement each other. The Chinese government has always attached great importance to higher education in ethnic minority areas. In Modernization of Education 2035, the development direction of improving the development level of higher education in ethnic minority areas is clearly proposed. At the same time, from a cultural perspective, the Chinese government is also committed to the common prosperity and development path of multi-ethnic cultural integration. Therefore, the authors believe that the regional unbalanced characteristics of higher education resources in China do not originate from different cultural differences. In essence, this core-margin unbalanced spatial pattern is the result of China’s unbalanced regional development strategy, which, since the reform and opening up, has prioritized the development of the eastern region. Therefore, the historical context has caused the spatial pattern of high-quality resources of higher education to be concentrated in the eastern region where those of Han nationality also cluster. We have mentioned this historical reality in the discussion section. Meanwhile, English language and style has been extensively edited.

Reviewer 2 Report

This research lacks a hypothesis. Even though the introduction states the need for this research, it does not clearly provide any mechanism to address the problem statement. Without this hypothesis, it is not possible to evaluate the  manuscript.

Author Response

Response 1: First, the basic hypothesis was proposed in section 2, line 159&160.

Therefore, we hypothesize that there must be geographical differences in the driving mechanisms behind the imbalanced characteristics of HED.

Second, four detail hypothesis were proposed according to index system in section 3. that is,

Hypothesis 1: Education input is a dominant factor that presents a negative spatial relationship with the development level of higher education; that is, the higher the level of higher education development in the region, the less restrictive the education input is in the region.

Hypothesis 2: Education access is an important institutional factor, and there is a negative spatial relationship in the development level of higher education, whereby the higher the level of HED in the region, the less restrictive the education access is in the region.

Hypothesis 3: The education process is the main factor restricting the high-quality development of higher education, and there is a positive spatial relationship with the development level of higher education, whereby the higher the level of HED in the region, the more restrictive the education access is in the region.

Hypothesis 4: Education outcome is the goal of higher education, and there is a positive spatial relationship with the development level of higher education, whereby the higher the level of HED in the region, the better the performance of education outcome is in the region.

Round 2

Reviewer 2 Report

This research needs a research question or hypothesis. This revised manuscript still lacks one.